# The global burden of age-related hearing loss among individuals aged 60 years and older: An analysis for the global burden of disease study 2021 and predictions to 2050

Peng Zhou[1]◉, Huiqin Wu[2]◉, Kui Xu[2], Xin Pan[2], Ling Li[2]*

1 Department of Otorhinolaryngology, Head and Neck Surgery, Zhongnan Hospital of Wuhan University, Wuhan, Hubei, China, 2 Department of Nuclear Medicine, Zhongnan Hospital of Wuhan University, Wuhan, Hubei, China

◉ Peng Zhou and Huiqin Wu contributed equally to this work and are co-first authors
* linglizn21@whu.edu.cn

## Abstract

### Objectives

Age-related hearing loss (ARHL) impacts communication, cognitive function, and mental health, with regional prevalence variation. We analyzed global spatial and temporal trends in ARHL burden from 1990 to 2021.

### Methods

Data on ARHL prevalence and years lived with disability (YLDs) among individuals aged 60 and older were analyzed. Estimated annual percentage changes (EAPC) in the age-standardized prevalence (ASPR) and YLD rates (ASYR) were analyzed by country, gender and socio-demographic index (SDI). Age-period-cohort, decomposition analyses, and frontier analysis identified influencing factors and quantify inequalities. Future trends were predicted using Bayesian model.

### Results

From 1990 to 2021, global ARHL prevalence cases increased by 137.43% (302.79 million to 718.93 million), with a 4.34% rise in ASPR. YLDs grew by 144.3% (10.76 million to 26.30 million), and ASYR rose by 4.45%. Males exhibited higher ASPR and ASYR despite more cases among females. High-middle and middle SDI regions showing higher and steadily increasing ASPR and ASYR, while high SDI regions exhibited the lowest and most stable rates. Population growth was the primary driver of the increasing burden. Projections indicate continued growth in ARHL cases by 2050.

**Data availability statement:** All relevant data are within the manuscript and its Supporting Information files.

**Funding:** This work is supported by the National Natural Science Foundation of China (82301789), the Program of Excellent Doctoral (Postdoctoral) of Zhongnan Hospital of Wuhan University (Grant No. ZNYB2021020), Youth Interdisciplinary Special Fund of Zhongnan Hospital of Wuhan University (Grant No. ZNQNJC2023007).

**Competing interests:** NO authors have competing interests.

## Conclusions

The ARHL burden among older adults has increased significantly, with marked regional and socio-demographic disparities. Lower SDI regions bear disproportionately higher burdens. Effective strategies, including public education, hearing protection, early screening, and research into treatments, are urgently needed to address this public health challenge.

## Introduction

Hearing loss has become a significant global public health concern, affecting an estimated 1.57 billion people worldwide in 2019, or approximately one in five individuals[1]. As the third leading cause of global years lived with disability (YLDs) and the primary cause of YLDs in adults over 70 [1], hearing loss profoundly impacts quality of life, including impaired communication, social interactions [2], and psychological effects such as loneliness, depression, and anxiety [3,4]. It also has socioeconomic consequences, such as increased unemployment rates [5]. In children, hearing loss hampers language development [6], while in older adults, it is associated with increased risk of cognitive decline [7].

Age-related hearing loss (ARHL), or presbycusis, is a progressive, symmetrical sensorineural impairment caused by cumulative aging effects on the auditory system. It affects approximately 50% of individuals in their seventh decade and is the predominant form of hearing loss in older adults [8]. ARHL prevalence varies globally: in the Netherlands, 33% of men and 31% of women over 65 have hearing loss of 35 dB or more [9], while in Europe, 30% of men and 20% of women at age 70 experience hearing loss, increasing to 55% and 45%, respectively, by age 80 [10]. In Asia, prevalence rates differ, with 45% of middle-aged and elderly adults affected in China [11] and 8% unilateral and 5.9% bilateral hearing loss reported in Korea [12].

Despite the substantial burden of ARHL on older populations, comprehensive analyses of global trends and determinants remain limited. Recently, Dong et al. [13] examined global ARHL trends from 1990–2021 using disability-adjusted life years (DALYs), providing valuable insights into the overall ARHL burden across all age groups. However, given that ARHL predominantly affects older adults and its impact is primarily on quality of life rather than mortality, there is a need for more targeted analyses focusing specifically on the older population and their disability burden.

The Global Burden of Diseases, Injuries, and Risk Factors Study (GBD) represents the most comprehensive worldwide observational epidemiological study to date, initiated in 1991 and coordinated by the Institute for Health Metrics and Evaluation (IHME) at the University of Washington [14]. This massive international collaboration involves over 11,000 researchers from more than 160 countries, employing sophisticated statistical modeling approaches to synthesize data from multiple sources including vital registration systems, verbal autopsy studies, surveillance data, censuses, household surveys, disease registries, and published literature [15]. The GBD framework implements rigorous data standardization protocols, bias adjustment

techniques, and uncertainty quantification methods, utilizing advanced computational tools and mathematical models to address data quality variations and ensure comparability across diverse contexts. The GBD 2021 study provides a standardized framework to assess ARHL epidemiology across 204 countries and territories from 1990 to 2021. Utilizing this comprehensive dataset, our study aims to: (1) analyze global ARHL burden trends specifically among older adults; (2) quantify socioeconomic disparities in ARHL burden using inequality indices; (3) identify key drivers of changing ARHL patterns through decomposition analysis; and (4) project future trends to 2050. These analyses will inform evidence-based policies and resource allocation strategies to address the growing challenge of ARHL in aging populations worldwide.

## Methods

### Data source and definitions

This study utilized data from the 2021 GBD Study (http://ghdx.healthdata.org/gbd-results-tool), which estimated the burden of 371 diseases or injuries and 88 risk factors across 204 countries and territories [15]. For the use of deidentified data in GBD study, a waiver of informed consent has been approved by the University of Washington Institutional Review Board. We extracted numbers and rates (with 95% uncertainty intervals) on the prevalence and YLDs for ARHL in individuals aged 60 years and older between 1990 and 2021. Following GBD methodology, hearing loss was defined as a hearing threshold exceeding 20 dB, measured by pure tone average (PTA) across 0.5, 1, 2, and 4 kHz frequencies in the better-performing ear[1]. Auditory impairment is categorized into six levels: mild (20–34 dB), moderate (35–49 dB), moderately severe (50–64 dB), severe (65–79 dB), profound (80–94 dB), and complete (≥95 dB). The estimates were generated using DisMod-MR 2.1, a Bayesian meta-regression tool that modeled severity-specific prevalence while accounting for hearing aid coverage and underlying causes. For ARHL specifically, estimates were derived by subtracting hearing loss attributable to other identified causes (congenital birth defects, otitis media, and meningitis) from total hearing loss prevalence. Data were differentiated by gender, age, region and country. Socio-demographic Index (SDI), representing a comprehensive development status that exhibits a robust correlation with health outcomes, was also included. According to SDI level, countries were classified into five quintiles: Low, Low-middle, Middle, High-middle, and High SDI [14].

### Statistical analysis

We calculated age-standardized rates (ASRs) per 100,000 individuals for ARHL in those above 60 years using the formula: $ASR = \frac{\sum_{i=1}^{N} a_i W_i}{\sum_{i=1}^{N} W_i}$, where $a_i$ is the age-specific rate in the $i$th age group, $w_i$ is the population count for the $i$th group in the standard population and, $N$ is the total number of age groups.

We calculated the estimated annual percentage change (EAPC) in ASRs of prevalence and YLDs to assess the average trends from 1990 and 2021 [16]. We modeled the natural logarithm of ASR using the linear regression equation: $\gamma = \alpha + \beta x + \varepsilon$, where $\gamma$ represents ln (ASR) and x denotes the calendar year. Therefore: ln (ASR) = $\alpha + \beta x + \varepsilon$, EAPC = 100 × ($e^{\beta} - 1$). The EAPC value indicates the direction and magnitude of temporal trends: EAPC > 0 represents an increasing trend, EAPC < 0 indicates a decreasing trend, and EAPC = 0 suggests no change. The absolute value of EAPC reflects the rate of change per year. Trends were classified as statistically significant increasing or decreasing if the EAPC 95% CI excluded zero, otherwise deemed statistically insignificant.

An age-period-cohort (APC) model was applied to evaluate ARHL burden by age, period, and birth cohort [17]. The age effect represents the social and biological processes of ageing. The period effect reflects temporal events influencing prevalence and YLDs rates in all age groups. The cohort effects refer to changes in disease burden that are attributable to varying degrees of risk factor exposure among different generations of the population. Data were organized into 5-year age intervals (1992–2021) and 12 overlapping 10-year birth cohorts (1892–1961). The APC model measures general and age-specific temporal trends as annual percentage changes in disease burden, reflecting calendar time and successive birth effects. Age effects are age-specific rates adjusted for periods, while period and cohort effects are relative risks

compared to a reference point. Age-period-cohort (APC) analysis was conducted using the online Age-Period-Cohort Analysis Tool provided by the National Institutes of Health (https://analysistools.cancer.gov/apc/) [18].

To analyze changes in ARHL prevalence and YLDs between 1990 and 2021, decomposition analyses examined the effects of population size, age distribution, and epidemiological trends [19]. Frontier analysis quantified the lowest potentially achievable age-standardized prevalence and YLDs rates by development status (SDI) [20].

The slope index of inequality and the concentration index measured the absolute and relative gradient inequality in ARHL across countries [21]. The slope index of inequality was calculated by regressing national YLDs rates against an SDI-linked relative position scale, based on the cumulative population midpoint. A weighted regression model was used to address heteroskedasticity [22].

Future ARHL burden (2022–2050) was predicted using a Bayesian APC model with integrated nested Laplacian approximation (R packages BAPC and INLA) [23], incorporating GBD Population Forecasts 2017–2100 for population projections [24]. This approach ensured robust predictions of disease burden across development levels.

All statistical analyses were performed using R statistical software (version 4.3.3), unless otherwise specified. Data visualization was conducted using the ggplot2 package and related packages within the R environment. Results

## Global trends

Overall, there were 302,792.64 thousand estimated prevalent cases of ARHL in people aged ≥60 years (95% UI 272,985.26−335,357.96) in 1990 and 718,929.06 thousand prevalent cases (95% UI 651042.00–788732.49) in 2021, with an increase of 137.43% (Table 1). The global ASPR rose from 63,481.94 per 100,000 in 1990–66,238.16 per 100,000 in 2021, representing a total growth of 4.34% with an EAPC of 0.12 (95% CI 0.11 to 0.13) (Table 1, Fig 1). While females consistently had higher case numbers than males, their ASPR was lower in both 1990 and 2021. Additionally, the trend of the EAPC showed a greater increase in females (EAPC = 0.15, 95% CI 0.13–0.16) compared to males (EAPC = 0.08, 95% CI 0.07–0.09) (Table 1, Fig 1).

The number of YLDs increased from 10,763.54 thousand (95% UI 7,056.52–15,718.28) in 1990 to 26295.53 thousand (95% UI 17,349.20–38,054.90) in 2021, with an increase of 144.3% (Table 2). The global ASYR grew from 2352.24 per 100,000 in 1990 to 2456.91 per 100,000 in 2021, representing a total growth of 4.45% with an EAPC of 0.17 (95% CI 0.15 to 0.19) (Table 2, Fig 1). While females had lower ASYR (1990: 2,242.40 per 100 000; 2021: 2,490.97 per 100 000) compared to males (1990: 2490.97 per 100 000; 2021: 2558.14 per 100 000), their YLDs were consistently higher (1990: 5808.30 thousand; 2021: 13921.89 thousand) than males (1990: 4955.24 thousand; 2021: 12373.64 thousand). The EAPC was also higher in females (0.19) than in males (0.14) (Table 2, Fig 1).

## SDI region level

Compared to 1990, the ASPR and ASYR in high-middle and middle SDI regions increased significantly in 2021, while other three regions (high, low-middle and low SDI) experienced only negligible increases or decreases (Table 1 and 2, Fig 1). In 2021, high SDI had the lowest ASPR (57650.42 per 100,000, 95% UI 52059.12–63889.02) and ASYR (2005.47 per 100,000 95% UI 1312.48–2931.31), while middle SDI had the highest ASPR (72365.56 per 100,000, 95% UI 65181.43–78912.01) and ASYR (2712.98 per 100,000, 95% UI 1794.50–3902.22). Notably, low SDI exhibited lower ASPR (61725.25 per 100,000, 95% UI 55749.18–68477.67) but relatively higher ASYR (2613.34 per 100,000, 95% UI 1741.73–3769.33) (Table 1 and 2).

## Regional trends

From 1990 to 2021, South Asia consistently reported the highest ASPR and the highest EAPC of 0.19. In 2021, its ASPR was 227,854.79 per 100,000 (95% UI 207238.24–252293.95). Southeast Asia and Oceania ranked second and third regions in ASPR, while Western Sub-Saharan Africa, Western Europe, and Southern Sub-Saharan Africa consistently

**Table 1. Prevalence and YLDs from ARHL in individuals aged 60 and older for 1990 and 2021, including estimated annual percentage changes from 1990 to 2021.**

| | Number | | | ASR | | | |
|---|---|---|---|---|---|---|---|
| | 1990 No.Í1000 (95% UI) | 2021 No.Í1000 (95% UI) | percentage change (100%) | 1990 per 100,000 (95% UI) | 2021 per 100,000 (95% UI) | percentage change (100%) | EAPC (95% CI) |
| Global | 302792.64 (272985.26-335357.96) | 718929.06 (651042.00-788732.49) | 137.43 | 63481.94 (57215.11-70275.92) | 66238.16 (59982.54-72669.82) | 4.34 | 0.12 (0.11 to 0.13) |
| Global_Female | 159837.92 (144038.26-177164.64) | 373573.10 (338159.75-410333.47) | 133.72 | 60337.55 (54363.45-66859.37) | 63650.49 (57619.38-69924.12) | 5.49 | 0.15 (0.13 to 0.16) |
| Global_Male | 142954.73 (128922.54-158176.94) | 345355.96 (312672.23-378669.99) | 141.58 | 67279.48 (60633.46-74367.81) | 69157.43 (62608.42-75808.62) | 2.79 | 0.08 (0.07 to 0.09) |
| High SDI | 82299.55 (74337.27-91195.20) | 161187.85 (145514.64-178587.79) | 95.86 | 56897.62 (51387.95-63027.01) | 57650.42 (52059.12-63889.02) | 1.32 | 0.02 (0.01 to 0.04) |
| High-middle SDI | 78659.28 (70836.51-87184.63) | 177257.06 (160249.06-193158.40) | 125.35 | 63726.06 (57374.45-70585.29) | 69115.59 (62494.18-75340.64) | 8.46 | 0.28 (0.26 to 0.3) |
| Middle SDI | 83203.17 (74587.36-92238.90) | 237804.55 (214108.66-259427.05) | 185.81 | 70771.12 (63453.02-78312.47) | 72365.56 (65181.43-78912.01) | 2.25 | 0.06 (0.05 to 0.08) |
| Low-middle SDI | 43241.76 (39160.95-47849.32) | 108094.78 (97956.10-119864.64) | 149.98 | 64194.22 (58099.75-71093.27) | 64439.66 (58368.22-71468.27) | 0.38 | −0.04 (−0.05 to −0.02) |
| Low SDI | 15051.45 (13564.13-16774.74) | 33980.42 (30672.37-37743.23) | 125.76 | 61027.17 (55030.65-67915.93) | 61725.25 (55749.18-68477.67) | 1.14 | 0.02 (−0.01 to 0.06) |
| Andean Latin America | 1257.69 (1135.54-1407.44) | 3854.34 (3478.06-4317.32) | 206.46 | 53806.04 (48534.89-60296.91) | 53779.08 (48509.47-60269.11) | −0.05 | 0.02 (0.01 to 0.04) |
| Australasia | 1916.72 (1800.69-2033.94) | 4583.52 (4092.38-5072.57) | 139.13 | 62008.73 (58215.47-65817.83) | 63946.44 (57048.39-70861.62) | 3.12 | 0.14 (0.09 to 0.2) |
| Caribbean | 1898.50 (1697.94-2142.39) | 4006.21 (3590.89-4518.46) | 111.02 | 59668.27 (53342.59-67332.21) | 59546.00 (53374.78-67160.33) | −0.2 | 0.01 (0 to 0.01) |
| Central Asia | 3349.12 (3004.57-3757.37) | 5763.02 (5188.01-6433.98) | 72.08 | 60755.37 (54470.58-68252.24) | 61069.36 (54920.38-68344.64) | 0.52 | 0.02 (0.02 to 0.02) |
| Central Europe | 11731.93 (10544.86-13150.80) | 18608.80 (16720.26-20859.94) | 58.62 | 61248.23 (55039.20-68692.93) | 61494.69 (55261.58-68891.41) | 0.4 | 0.02 (0.01 to 0.02) |
| Central Latin America | 5666.19 (5089.36-6390.55) | 18392.59 (16470.25-20710.87) | 224.6 | 60190.92 (54012.93-67942.50) | 60077.80 (53775.54-67701.04) | −0.19 | 0.01 (0 to 0.01) |
| Central Sub-Saharan Africa | 1263.04 (1143.67-1391.96) | 2932.37 (2658.22-3230.29) | 132.17 | 53351.19 (48401.13-58846.06) | 52856.66 (48008.18-58219.66) | −0.93 | −0.02 (−0.03 to −0.01) |
| East Asia | 78838.58 (69777.97-88200.38) | 227625.34 (202950.99-247007.62) | 188.72 | 77287.38 (68465.32-86072.98) | 81673.54 (72888.25-88654.95) | 5.68 | 0.19 (0.16 to 0.21) |
| Eastern Europe | 22049.73 (19845.82-24685.25) | 29567.33 (26616.70-33067.05) | 34.09 | 61492.43 (55305.15-68888.97) | 61782.38 (55606.55-69145.29) | 0.47 | 0.02 (0.02 to 0.02) |
| Eastern Sub-Saharan Africa | 5610.70 (4891.55-6445.37) | 12723.65 (11108.21-14486.50) | 126.77 | 68980.28 (60227.29-78817.97) | 70984.08 (62073.62-80441.83) | 2.9 | 0.1 (−0.02 to 0.22) |
| High-income Asia Pacific | 14189.67 (12867.50-15815.86) | 35274.80 (31831.91-39422.20) | 148.59 | 56801.95 (51461.19-63331.92) | 57043.78 (51620.28-63584.02) | 0.43 | 0.02 (0.02 to 0.03) |
| High-income North America | 29741.67 (26296.57-33297.43) | 54210.37 (48181.26-60826.15) | 82.27 | 63332.26 (56002.81-70924.74) | 60910.78 (54137.41-68339.02) | −3.82 | −0.12 (−0.15 to −0.09) |
| North Africa and Middle East | 21056.04 (19028.68-23335.40) | 57491.18 (51956.04-63660.05) | 173.04 | 57608.53 (52046.43-63892.33) | 57614.70 (52050.35-63831.05) | 0.01 | 0 (0–0) |
| Oceania | 225.30 (199.15-254.70) | 567.90 (502.04-642.69) | 152.06 | 71997.14 (63636.33-81225.95) | 73094.40 (64625.54-82595.76) | 1.52 | −0.04 (−0.07 to −0.01) |
| South Asia | 80644.00 (73268.70-89219.43) | 227854.79 (207238.24-252293.95) | 182.54 | 64835.64 (58834.41-71861.14) | 65249.36 (59293.58-72311.05) | 0.64 | −0.05 (−0.07 to −0.03) |

*(Continued)*

**Table 1.** (Continued)

| | Number | | | ASR | | | |
|---|---|---|---|---|---|---|---|
| | 1990 No.Í1000 (95% UI) | 2021 No.Í1000 (95% UI) | percentage change (100%) | 1990 per 100,000 (95% UI) | 2021 per 100,000 (95% UI) | percentage change (100%) | EAPC (95% CI) |
| Southeast Asia | 21069.81 (18849.00-23524.17) | 58613.97 (52310.35-65568.72) | 178.19 | 73966.73 (66146.86-82507.29) | 75023.62 (66942.44-83821.58) | 1.43 | −0.03 (−0.05 to 0) |
| Southern Latin America | 3217.86 (2877.96-3601.11) | 6281.81 (5628.25-7048.79) | 95.22 | 55385.07 (49489.60-62025.90) | 55426.44 (49670.96-62169.46) | 0.07 | 0.01 (0 to 0.01) |
| Southern Sub-Saharan Africa | 1556.51 (1429.47-1696.72) | 3373.70 (3109.33-3660.96) | 116.75 | 50084.21 (45991.62-54635.91) | 50423.69 (46459.65-54760.10) | 0.68 | 0.02 (0 to 0.04) |
| Tropical Latin America | 7116.04 (6332.95-8030.10) | 21622.23 (19270.82-24382.74) | 203.85 | 67741.19 (60259.88-76362.33) | 67566.61 (60206.44-76169.16) | −0.26 | −0.01 (−0.02 to 0.01) |
| Western Europe | 36400.50 (33122.81-40026.17) | 58506.77 (53184.64-64386.29) | 60.73 | 47328.46 (43081.54-52001.00) | 47301.67 (43052.38-51970.41) | −0.06 | −0.05 (−0.06 to −0.03) |
| Western Sub-Saharan Africa | 4843.07 (4401.57-5319.20) | 9747.34 (8864.65-10684.32) | 101.26 | 48957.01 (44590.78-53676.76) | 46704.51 (42598.79-51077.21) | −4.6 | −0.11 (−0.19 to −0.03) |

ARHL, age-related hearing loss.

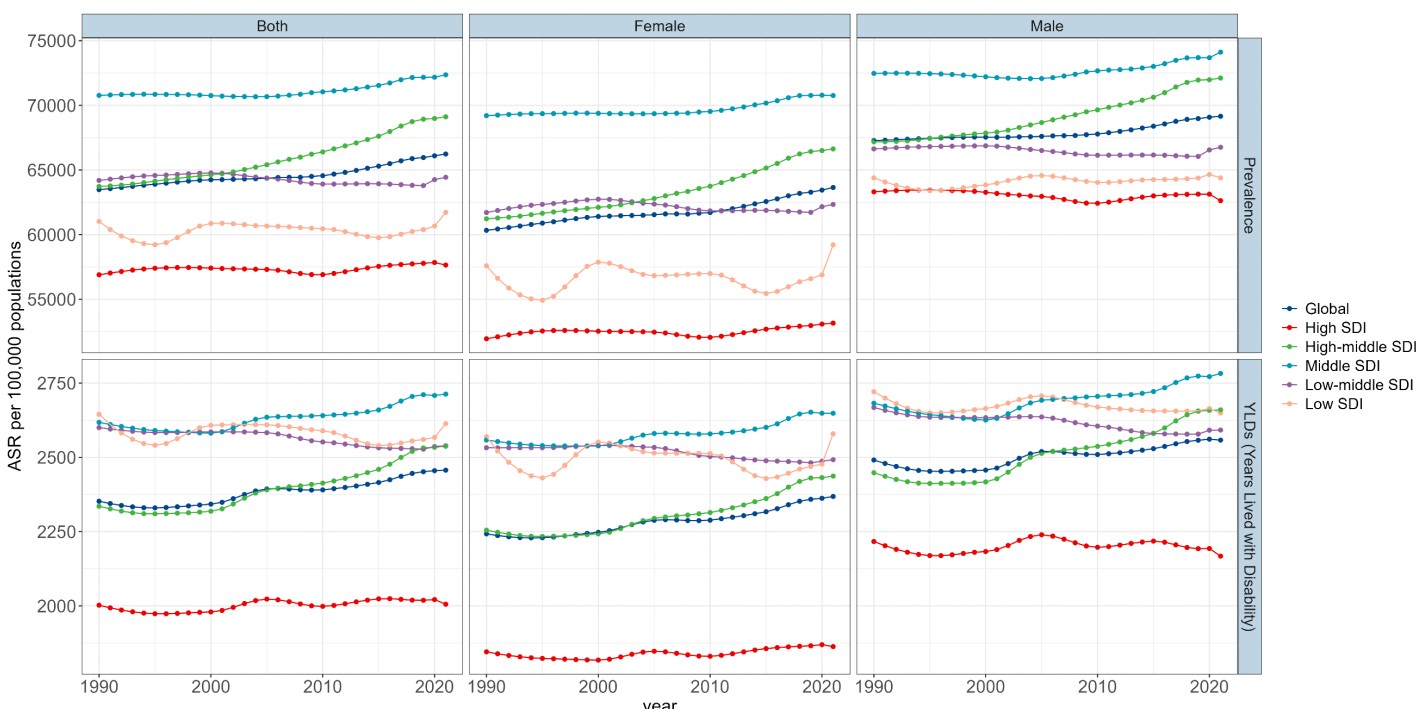

**Fig 1. Burden of ARHL trends in global and five SDI regions in individuals aged 60 and older, from 1990 to 2021.** SDI, Socio-demographic Index; ASIR, age-standardized incidence rate; YLDs, years lived with disability; ARHL, age-related hearing loss.

**Table 2. YLDs from ARHL in individuals aged 60 and older for 1990 and 2021, including estimated annual percentage changes from 1990 to 2021.**

| | Number | | | ASR | | | |
|---|---|---|---|---|---|---|---|
| | 1990 No.Í1000 (95% UI) | 2021 No.Í1000 (95% UI) | percentage change (100%) | 1990 per 100,000 (95% UI) | 2021 per 100,000 (95% UI) | percentage change (100%) | EAPC (95% CI) |
| Global | 10763.54 (7056.52-15718.28) | 26295.53 (17349.20-38054.90) | 144.3 | 2352.24 (1555.83-3412.61) | 2456.91 (1626.46-3547.91) | 4.45 | 0.17 (0.15 to 0.19) |
| Global_Female | 5808.30 (3810.36-8469.26) | 13921.89 (9219.77-20107.16) | 139.69 | 2242.40 (1477.72-3258.05) | 2368.34 (1568.25-3421.53) | 5.62 | 0.19 (0.18 to 0.21) |
| Global_Male | 4955.24 (3240.40-7243.29) | 12373.64 (8124.75-17998.78) | 149.71 | 2490.97 (1652.05-3603.92) | 2558.14 (1692.33-3702.71) | 2.7 | 0.14 (0.11 to 0.16) |
| High SDI | 2894.35 (1902.06-4220.61) | 5796.56 (3819.72-8431.99) | 100.27 | 2002.16 (1315.50-2919.59) | 2005.47 (1312.48-2931.31) | 0.17 | 0.07 (0.05 to 0.09) |
| High-middle SDI | 2743.98 (1788.86-4013.56) | 6436.66 (4224.97-9307.79) | 134.57 | 2335.18 (1537.58-3390.62) | 2537.59 (1669.62-3664.37) | 8.67 | 0.32 (0.29 to 0.36) |
| Middle SDI | 2866.37 (1863.23-4202.24) | 8617.40 (5654.28-12456.62) | 200.64 | 2617.92 (1729.72-3793.92) | 2712.98 (1794.50-3902.22) | 3.63 | 0.15 (0.12 to 0.17) |
| Low-middle SDI | 1642.25 (1080.30-2407.92) | 4069.41 (2690.84-5946.65) | 147.79 | 2600.30 (1736.67-3767.82) | 2539.14 (1696.94-3680.02) | −2.35 | −0.09 (−0.11 to −0.08) |
| Low SDI | 604.58 (396.10-884.95) | 1353.39 (889.33-1971.33) | 123.86 | 2645.14 (1764.93-3819.82) | 2613.34 (1741.73-3769.33) | −1.2 | −0.03 (−0.07 to 0.01) |
| Andean Latin America | 43.54 (28.17-64.48) | 135.08 (87.56-199.38) | 210.27 | 1924.66 (1252.05-2839.27) | 1907.36 (1238.68-2811.17) | −0.9 | 0.06 (0.02 to 0.1) |
| Australasia | 64.62 (42.88-93.33) | 156.60 (101.44-230.96) | 142.34 | 2119.19 (1409.72-3056.43) | 2133.04 (1374.55-3157.76) | 0.65 | 0.21 (0.11 to 0.31) |
| Caribbean | 68.34 (44.35-100.74) | 146.95 (95.82-215.98) | 115.02 | 2199.09 (1434.18-3229.93) | 2173.75 (1416.00-3196.64) | −1.15 | −0.01 (−0.02 to 0) |
| Central Asia | 119.59 (77.25-176.33) | 197.26 (127.42-291.96) | 64.95 | 2249.71 (1461.80-3303.89) | 2235.65 (1463.94-3279.79) | −0.62 | 0.02 (0 to 0.04) |
| Central Europe | 419.30 (273.23-618.72) | 689.70 (451.20-1012.98) | 64.49 | 2274.21 (1492.91-3337.97) | 2256.57 (1474.12-3317.39) | −0.78 | 0.03 (0.01 to 0.05) |
| Central Latin America | 198.92 (129.16-293.76) | 654.99 (426.67-966.03) | 229.27 | 2202.94 (1442.07-3233.42) | 2181.92 (1426.77-3208.79) | −0.95 | 0.01 (−0.01 to 0.02) |
| Central Sub-Saharan Africa | 51.62 (33.16-76.63) | 120.26 (78.36-176.79) | 132.96 | 2421.04 (1595.51-3521.80) | 2374.14 (1575.41-3436.74) | −1.94 | −0.04 (−0.06 to −0.02) |
| East Asia | 2537.27 (1626.10-3742.97) | 8077.28 (5283.23-11654.03) | 218.35 | 2710.78 (1768.71-3945.20) | 2975.02 (1956.46-4275.56) | 9.75 | 0.38 (0.32 to 0.44) |
| Eastern Europe | 814.89 (535.32-1193.06) | 1104.96 (725.35-1610.19) | 35.6 | 2362.54 (1562.05-3441.06) | 2339.66 (1540.29-3404.65) | −0.97 | 0 (−0.01 to 0.01) |
| Eastern Sub-Saharan Africa | 233.94 (149.52-345.34) | 530.92 (340.95-782.93) | 126.94 | 3110.71 (2028.92-4521.70) | 3164.38 (2063.82-4608.08) | 1.73 | 0.06 (−0.07 to 0.19) |
| High-income Asia Pacific | 450.83 (292.42-670.53) | 1233.95 (813.37-1806.52) | 173.71 | 1860.07 (1213.49-2753.23) | 1837.27 (1192.13-2722.78) | −1.23 | 0.06 (0.03 to 0.1) |
| High-income North America | 1139.83 (746.66-1659.92) | 2057.91 (1355.47-2999.93) | 80.54 | 2410.51 (1575.72-3516.13) | 2296.28 (1510.68-3350.69) | −4.74 | −0.11 (−0.16 to −0.06) |
| North Africa and Middle East | 876.33 (581.80-1271.03) | 2315.43 (1525.24-3384.18) | 164.22 | 2529.75 (1703.47-3635.51) | 2420.22 (1613.11-3510.11) | −4.33 | −0.11 (−0.13 to −0.1) |
| Oceania | 6.89 (4.34-10.35) | 17.58 (11.14-26.28) | 155.06 | 2489.10 (1613.94-3665.85) | 2497.94 (1618.89-3679.77) | 0.35 | −0.03 (−0.05 to −0.01) |
| South Asia | 3074.82 (2023.60-4499.94) | 8613.61 (5718.62-12529.93) | 180.13 | 2655.14 (1776.26-3833.51) | 2583.17 (1732.46-3728.59) | −2.71 | −0.12 (−0.13 to −0.1) |

*(Continued)*

**Table 2.** (Continued)

| | Number | | | ASR | | | |
|---|---|---|---|---|---|---|---|
| | 1990 No.Í1000 (95% UI) | 2021 No.Í1000 (95% UI) | percentage change (100%) | 1990 per 100,000 (95% UI) | 2021 per 100,000 (95% UI) | percentage change (100%) | EAPC (95% CI) |
| Southeast Asia | 743.41 (483.22-1096.01) | 2034.96 (1324.14-3004.30) | 173.73 | 2775.20 (1831.81-4049.61) | 2756.42 (1819.64-4032.65) | −0.68 | −0.07 (−0.09 to −0.06) |
| Southern Latin America | 113.68 (73.81-166.76) | 227.52 (148.39-336.91) | 100.13 | 2018.31 (1318.70-2946.85) | 1987.75 (1293.90-2947.44) | −1.51 | −0.01 (−0.03 to 0) |
| Southern Sub-Saharan Africa | 56.36 (36.78-82.96) | 118.69 (77.24-173.50) | 110.6 | 1910.17 (1259.13-2788.88) | 1896.74 (1249.66-2746.26) | −0.7 | 0 (−0.03 to 0.02) |
| Tropical Latin America | 251.07 (163.26-370.34) | 782.90 (511.30-1145.36) | 211.83 | 2514.59 (1655.74-3675.51) | 2485.97 (1629.77-3626.96) | −1.14 | −0.04 (−0.08 to 0.01) |
| Western Europe | 1264.63 (827.23-1859.49) | 2129.33 (1398.77-3098.44) | 68.38 | 1626.54 (1062.08-2393.40) | 1611.32(1046.16-2364.01) | −0.94 | 0.06 (0.02 to 0.09) |
| Western Sub-Saharan Africa | 209.24 (137.17-303.30) | 414.19 (270.64-599.41) | 97.95 | 2227.17 (1480.11-3196.48) | 2097.54 (1390.29-3004.20) | −5.82 | −0.11 (−0.19 to −0.04) |

YLDs, years lived with disability; ARHL, age-related hearing loss

reported the lowest ASPR (Table 1, S1 Fig). For ASYR, Eastern Sub-Saharan Africa recorded the highest ASYR (3164.38 per 100,000; 95% UI 2063.82–4608.08) in 2021 and maintained the highest ASYR for most of the period. South Asia and Southeast Asia followed with the second and third highest ASYR. Corresponding with the ASPR results, Western Europe had the lowest ASYR (1611.32 per 100,000; 95% UI 2063.82–4608.08) (Table 2, S1 Fig).

## National trends

In 2021, China, India and the USA recorded the highest ARHL prevalence and YLDs among people aged ≥60 years, with China leading at 4.45 e+8 cases (4.217715e+08− 4.719349e+08) and 7.878745e+06 YLDs (5.151003e+06− 1.136675e+07), followed by India with 2.532398e+08 (2.416939e+08, 2.651484e+08) cases and 3.541291e+06 YLDs (2.356028e+06−5.149264e+06), and the USA with 7.168475e+07 (6.783383e+07−7.623416e+07) cases and 1.846633e+06 YLDs (1.214904e+06−2.684514e+06) as shown in (S1 Table,)

China reported the highest ASPR with 82162.49 cases per 100 000 individuals (95% UI 89187.21–73288.08), closely followed by Myanmar (75477.46, 67395.29–84528.14) and Philippines (75372.43, 67388.97–84075.94) (S1 Table). The United Arab Emirates and Qatar witnessed the largest increases in prevalence, with rises of 793.3% and a 784.3% between 1990 and 2021 (S1 Table).

For ASYR in 2021, Kenya had the highest rate with 3523.953 per 100,000 (95% UI 2307.0686–5091.372), followed by Madagascar (3431.491, 95% UI 2250.5826–4937.814) and Malawi (3318.573, 95% UI 2208.3219–4758.618) (S1 Table). Qatar saw a notable increase in YLDs of 727.3% over 30 years, while Niue experienced a reduction of 8.1% (S1 Table).

## Age, period and cohort effects

Between 1990 and 2021, the annual changes (% per year) of global prevalence and YLDs consistently remained above zero across all age groups (Fig 2A, Fig 3A). However, these rates of increase demonstrated a decreasing trend with advancing age, suggesting slower increases among older age groups. The annual changes in prevalence and YLDs in females were consistently higher than those in males, indicating a greater disease burden increase in females (Fig 2A, Fig 3A).

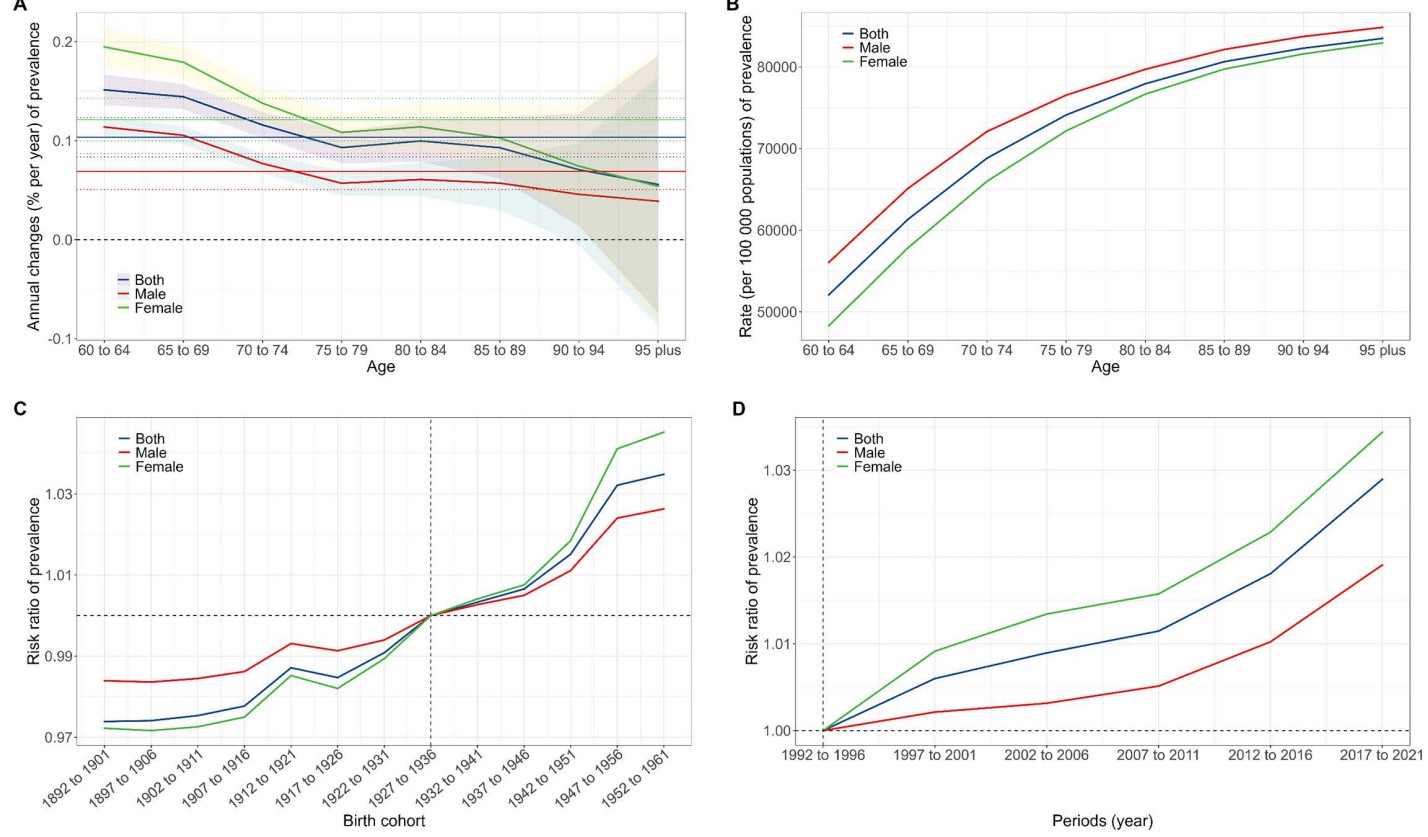

**Fig 2. Prevalence of ARHL among individuals aged 60 and older analyzed through age-period-cohort analysis from 1990 to 2021.** A: annual changes by age; B: age effect; C: cohort effect; D: period effect. ARHL, age-related hearing loss.

Prevalence and YLDs risks increased with age for both genders, though rates were consistently higher in males across all age groups (Fig 2B, Fig 3B). Among the 13 consecutive 10-year birth cohorts (1892–1961), cohort risks for prevalence and YLDs showed continuous increases (Fig 2C, Fig 3C). Period effects also indicated rising risks for prevalence and YLDs over time, affecting both genders (Fig 2D, Fig 3D).

## Decomposition and frontier analyses

Globally, across the five SDI regions and within the 21 geographical regions, the primary driver for the increase in the ARHL prevalence and the associated YLDs was population growth from 1990 to 2021, regardless of gender (Fig 4). In the High-income Asia Pacific region, the increase in ARHL prevalence and YLDs was notably attributable to aging in both genders, in addition to population growth, a trend more pronounced than in other regions (Fig 4).

We calculated the effective difference from the frontier for each country and territory using YLDs (1990–2021) and SDI (Fig 5). As SDI increases, both the effective difference and variance tend to decrease, indicating that countries or regions with lower SDI have greater potential to reduce the ARHL burden (Fig 5A). Kenya, Madagascar, Malawi, China and Ethiopia showed the largest deviation from the frontier, reflecting significantly higher YLDs compared to other nations with similar sociodemographic resources (Fig 5B). Countries and territories positioned on the efficiency frontier with low SDI (<0.5) and minimal effective differences included Niger, Chad, Mali, Burkina Faso, and Guinea. Conversely, those with high SDI

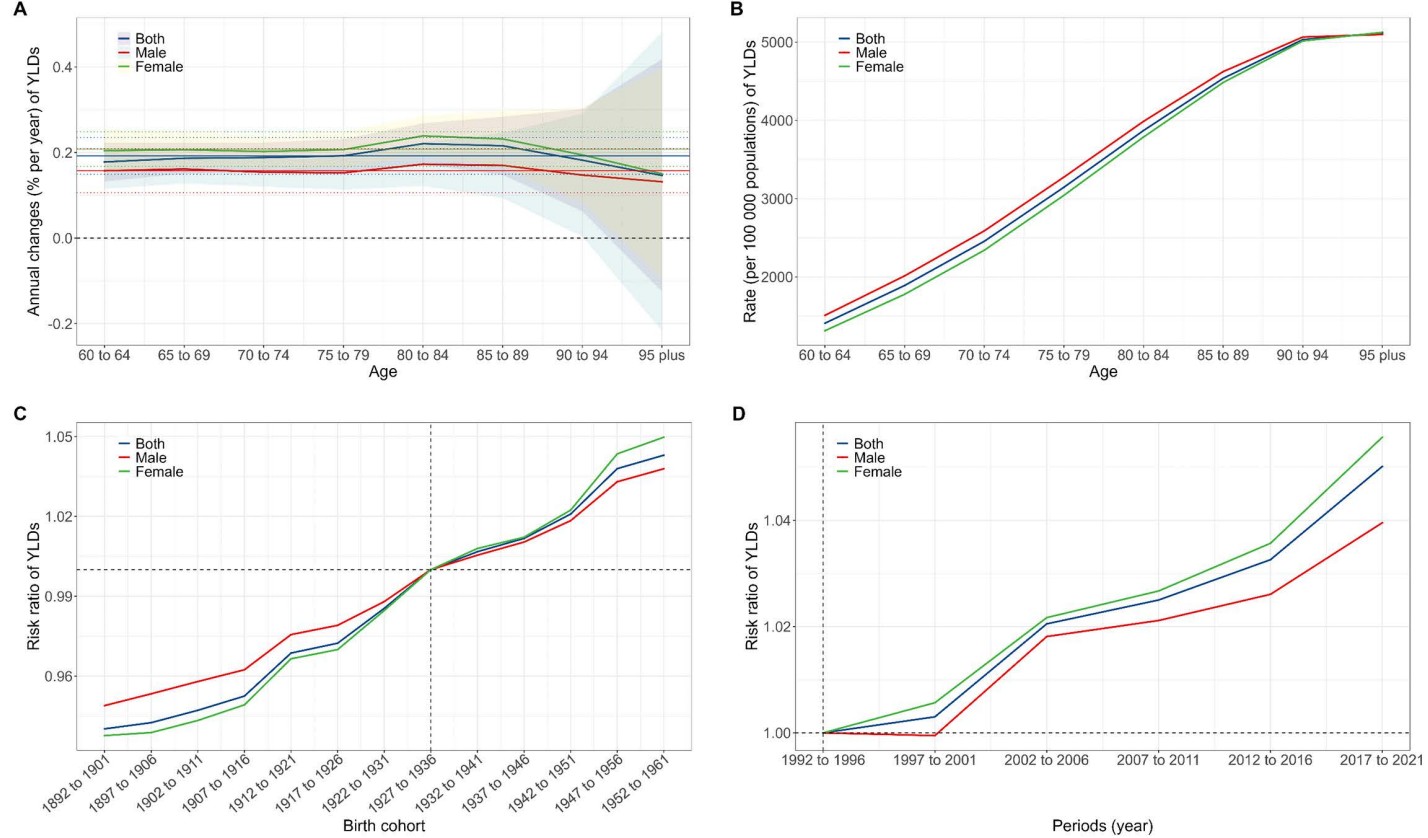

**Fig 3. YLDs of ARHL among individuals aged 60 and older analyzed through age-period-cohort analysis from 1990 to 2021.** A: annual changes by age; B: age effect; C: cohort effect; D: period effect. YLDs, years lived with disability; ARHL, age-related hearing loss.

(>0.85) demonstrating substantial effective differences relative to their developmental status comprised the USA, Japan, Canada, Lithuania, and Taiwan (Province of China).

### The slope index of inequality and projection from 2022 to 2050

The inequality slope index revealed that the disparity in ARHL prevalence and YLDs rates f between countries with the highest and lowest SDI shifted from −3265.4 and −514.1 in 1990 to −3373.34 and −320.69 in 2021. This change indicates that although the inequity in prevalence rates has slightly eased in countries with lower SDI, the inequality in YLDs has worsened (Fig 6).

Projections from 2022 to 2050 indicate that the ASPR globally will remain relatively stable in both sexes, the ARHL prevalence cases were projected to continue increasing among both females and males globally (Fig 7).

### Discussion

This study provides a comprehensive analysis of the global burden of ARHL among people aged 60 years and older from 1990 to 2021, highlighting significant geographical, gender, and sociodemographic variations in prevalence and YLDs. Between 1990–2021, prevalent ARHL cases more than doubled, increasing by 137.43%, from 302.79 million to 718.93 million, while the global ASPR rose by 4.34%. Similarly, YLDs grew by 144.3%, from 10.76 million to 26.30 million, with ASYR rising by 4.45%. Despite higher absolute numbers of cases and YLDs in females, males consistently exhibited

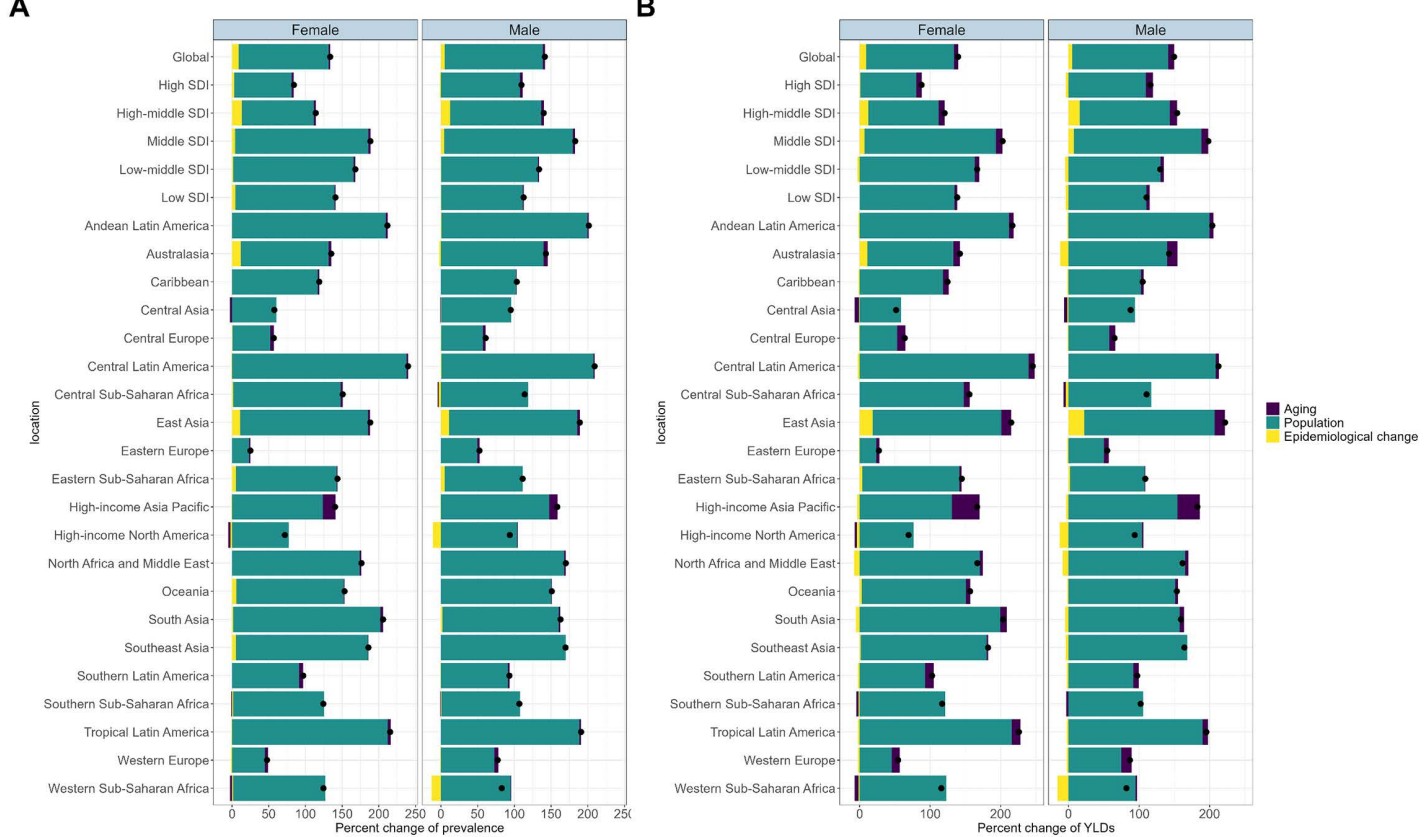

**Fig 4. Decomposition analysis of prevalence and YLDs of ARHL among individuals aged 60 and older by gender.** A black dot represents the overall change attributed to three components: aging, population growth, and epidemiological alterations. YLDs, years lived with disability; ARHL, age-related hearing loss.

higher ASPR and ASYR throughout the study period. Marked regional, national, and SDI disparities further underscored significant differences in ARHL burden worldwide. Age, period, and cohort analyses indicated that ARHL burden increased with advancing age, over time, and across successive birth cohorts in both genders. Population growth emerged as the primary driver of the rising global ARHL burden. Projections to 2050 suggest that while the global ASPR will remain stable, the number of ARHL cases continue to increase.

Our results revealed a substantial increase in absolute numbers of ARHL cases and YLDs, primarily driven by global population growth and aging, irrespective of gender. As life expectancy increases, more individuals reach ages where ARHL is prevalent. The modest increase in ASPR suggests additional underlying risk factors beyond demographic shifts alone, such as hypertension [25], diabetes mellitus [26], gout [27], smoking [28], and noise exposure [29]. The higher EAPC of prevalence and YLDs observed in females compared to males highlights a potential gender disparity in ARHL progression. This phenomenon may be partially attributed to the diminishing protective effect of estrogen on auditory function in older women due to age-related declines in endogenous estrogen levels [30]. Hormone-based therapies or gender-specific preventive strategies may be beneficial in addressing the accelerated progression of ARHL in women.

Our findings complement and extend the recent comprehensive analysis by Dong et al. [13], who examined ARHL burden across all age groups using the same GBD 2021 dataset. While Dong et al. reported 1.55 billion global ARHL

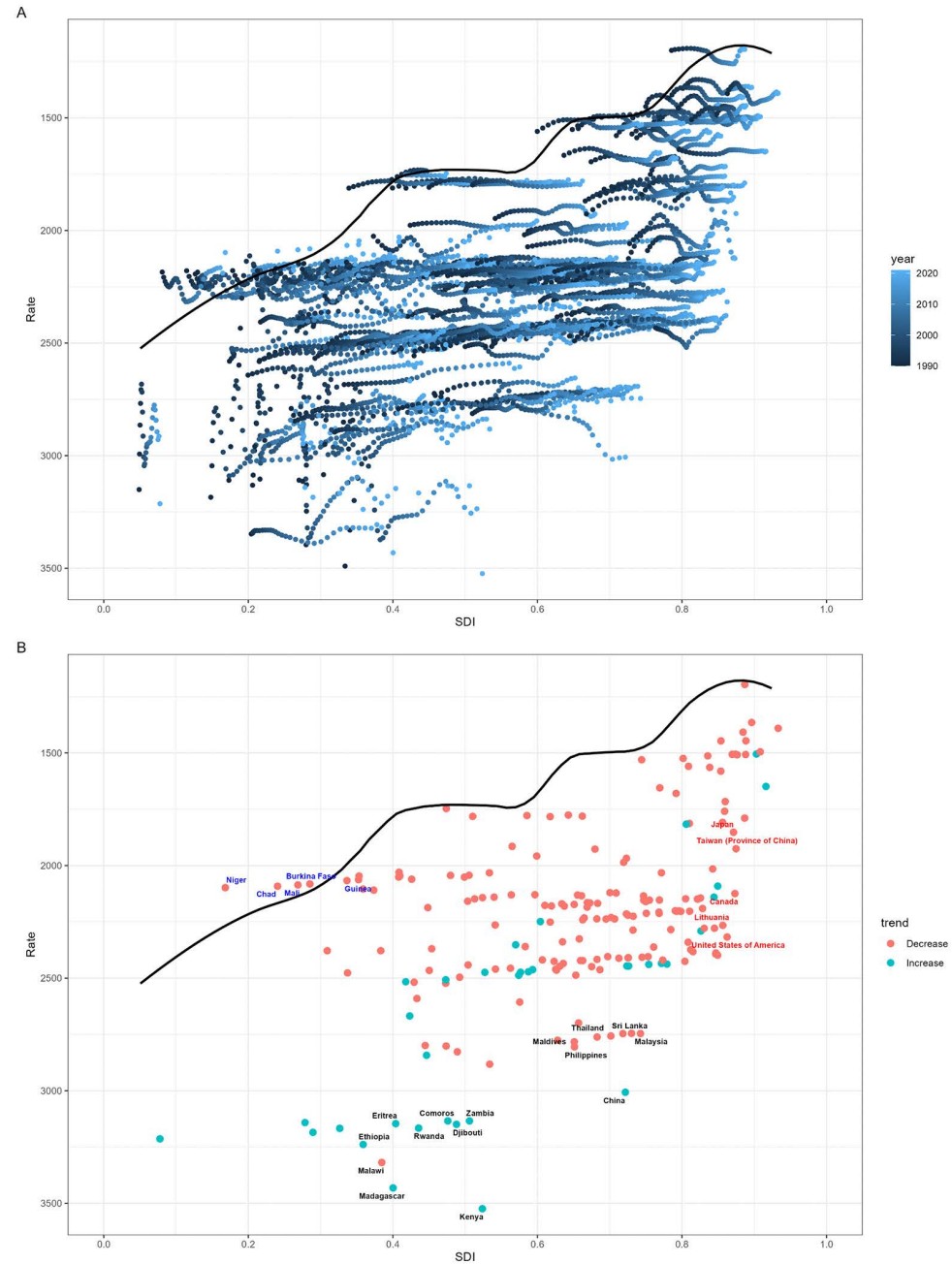

**Fig 5. Frontier analysis examining the relationship between SDI and ARHL burden among individuals aged 60 and older. (A)** Temporal progression from 1990 to 2021, with color gradients representing years (dark blue: 1990; light blue: 2021). The black line delineates the efficiency frontier. **(B)** Cross-sectional analysis for 2021, where each dot represents a country or territory. Countries with the largest effective differences (top 15) are labeled in black (e.g., Eritrea, Ethiopia); low-SDI frontier countries (<0.5) with minimal effective differences are shown in blue (e.g., Niger, Chad); high-SDI countries (>0.85) with relatively large effective differences are highlighted in red (e.g., USA, Japan). Dot colors indicate temporal changes in ARHL burden: red represents decreasing YLDs, blue represents increasing YLDs between 1990 and 2021.

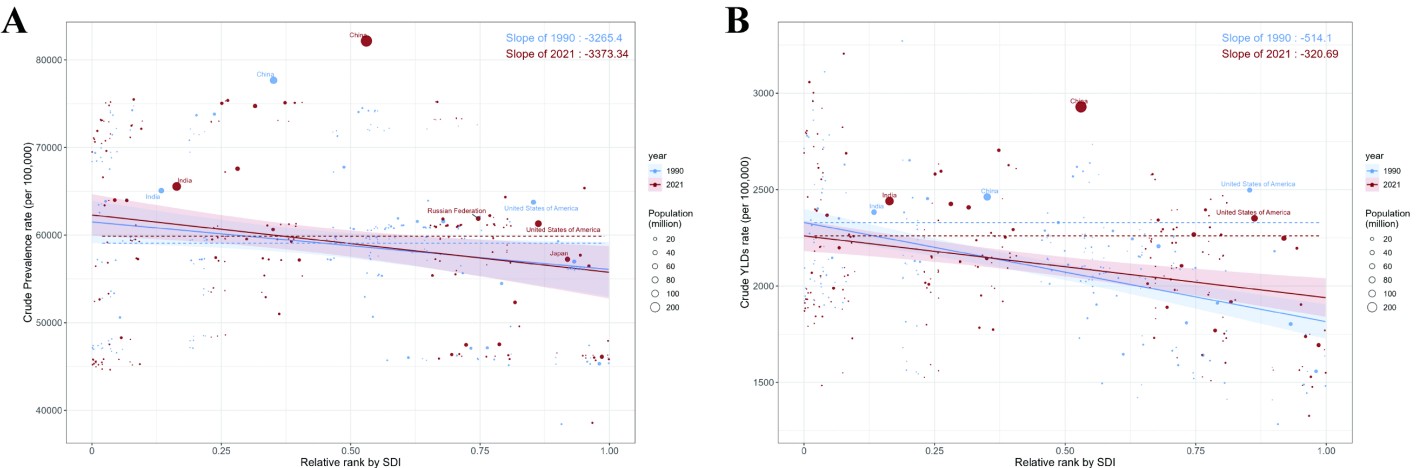

**Fig 6. Regression curves illustrating health inequalities associated with ARHL in terms of prevalence and YLDs.** ARHL, age-related hearing loss; YLDs, years lived with disability.

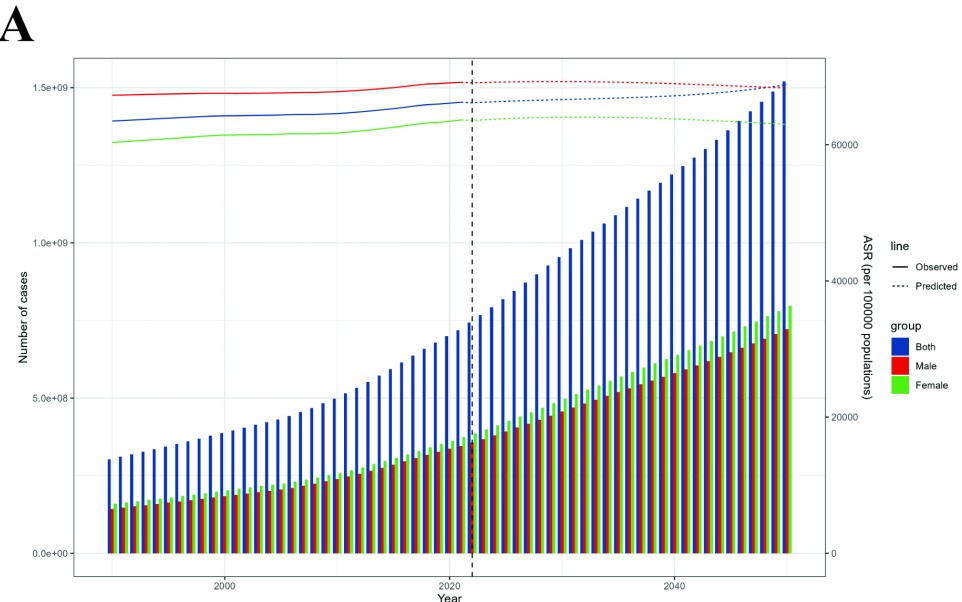

**Fig 7. Projection of the number of ARHL cases among individuals aged 60 and older from 2022 to 2050.** ARHL, age-related hearing loss.

cases across all ages, our focused analysis on the 60 + population reveals that this age group alone accounts for approximately 46% (718.93 million) of the total global burden. Both studies consistently identify middle SDI regions as bearing the highest ASPR and ASYR, with Dong et al. reporting an ASPR of 19,715.49 per 100,000 in middle SDI regions across all ages, while our elderly-specific analysis shows an even higher ASPR of 72,365.56 per 100,000 in the same regions. This convergence of findings across different analytical approaches strengthens the evidence that middle SDI regions face particular challenges in managing ARHL burden. Notably, while Dong et al. identified the 55–69 age group as having the

highest absolute prevalence, our decomposition analysis reveals that population growth, rather than aging alone, drives the burden increase in the 60 + population, suggesting that preventive interventions should begin before age 60.

High-middle and middle SDI regions exhibited higher ASPR and ASYR for ARHL, with a consistent upward trajectory over the years, aligning with previous findings [31] and corroborating Dong et al.'s observations across all age groups [13]. Such trends may be attributed to several factors, including lifestyle changes, rapid industrialization and urbanization leading to increased noise pollution, and improvements in diagnostic capabilities. From a health policy perspective, these consistent patterns across studies have important implications for resource allocation. Middle SDI countries experiencing rapid industrialization require immediate implementation of: (1) mandatory occupational noise exposure limits following WHO recommendations, (2) urban planning regulations that consider acoustic environments, and (3) subsidized hearing aid programs targeting populations aged 50 and above, rather than waiting until age 60. For low-middle and low SDI regions, where our analysis shows relatively stable but high burden, community-based screening programs using mobile health technologies and training of community health workers represent cost-effective starting points. Therefore, cost-effective interventions such as hearing screening for older adults, hearing technologies (e.g., hearing aids, cochlear implants) and clinical management of ear diseases, are crucial in addressing this issue [32–34]. Conversely, high SDI regions maintained the lowest and most stable ASPR and ASYR, likely due to earlier adoption of preventive measures, stricter noise regulations, broader public awareness of hearing protection, and enhanced access to high-quality auditory healthcare [31,35].

Our inequality analysis reveals critical health equity dimensions not explored in previous studies. The slope index of inequality demonstrated that while prevalence disparities between high and low SDI countries remained relatively stable (−3265.4 to −3373.34), YLD inequalities actually improved (−514.1 to −320.69), suggesting that lower SDI countries may be making progress in managing ARHL-related disability despite persistent prevalence gaps. This finding, unique to our YLD-focused approach, indicates that interventions in lower SDI regions may be effectively reducing disability severity even when prevalence remains high. In contrast, Dong et al.'s DALY-based analysis showed consistent disparities without capturing this nuanced improvement in disability management.

Consistent with SDI trends, South Asia reported the highest ASPR and the largest EAPC, aligning with the overall higher ASPR observed in middle SDI regions. Conversely, Western Europe maintained the lowest ASPR and ASYR, reflecting trends in high SDI regions, likely due to better healthcare access and preventive measures. Nationally, China, India, and the United States ranked highest in absolute ARHL cases and YLDs, underscoring the challenges faced by these populous nations while also highlighting intra-SDI variability.

The age-period-cohort analysis of annual changes in ARHL prevalence and YLDs from 1990 to 2021 reveals a consistent increase across all age groups, underscoring the growing burden of hearing loss worldwide. The age effect analysis confirms the well-established relationship between ageing and higher ARHL prevalence and YLDs. The consistently higher rates observed in males align with previous research suggesting greater susceptibility to hearing loss in men [36], possibly due to factors such as occupational noise exposure or genetic predisposition [37]. The cohort effect analysis reveals a continuous increase in ARHL risk, potentially driven by cumulative exposure to risk factors, intergenerational lifestyle and occupational changes, or increased longevity. The observed period effects, showing a rising risk for ARHL over time for both genders, suggest the influence of broader societal and environmental changes, such as increased noise pollution, changes in healthcare practices, or shifts in the recognition and reporting of hearing loss.

The frontier analysis revealed an inverse relationship between SDI and the effective difference from the frontier, with higher SDI countries generally showing smaller effective differences and less variance. This suggests that as countries develop socioeconomically, they tend to converge towards more efficient management of ARHL burden relative to their resources. However, the identification of countries like Kenya, Madagascar, Malawi, China, and Ethiopia as having the highest effective difference from the frontier underscores significant opportunities for improvement in ARHL management in these nations, despite their varying SDI levels.

Our analysis of the slope index of inequality highlights global health disparities in ARHL. The slope index of inequality for ARHL prevalence remained relatively stable from 1990 to 2021, while the slope index for YLDs increased, suggesting reduced inequality. The decreased ARHL burden in lower SDI countries may be attributed to improved access to fundamental audiological care and the implementation of early intervention strategies. The increased ARHL burden in higher SDI countries likely reflects the demographic transition associated with aging, a key risk factor for ARHL.

Projections for 2022–2050 indicate a stabilization in global ASPR for both genders, despite a continued increase in absolute ARHL prevalence cases driven by population growth and aging. This highlights the need for proactive policies and interventions to address the anticipated increase in ARHL burden, particularly in regions experiencing rapid demographic changes or industrialization and urbanization.

However, some limitations should be noted. Firstly, the ARHL data within the 2021 GBD study, derived from secondary sources, inherently contained biases and gaps affecting the ARHL burden estimates. The reliance on modeled data is particularly problematic for countries with limited primary audiometric surveillance data. Policymakers in data-sparse regions should interpret absolute burden estimates with caution while focusing on the relative trends and patterns. Secondly, variations in the diagnosis and detection protocols for ARHL across different countries and time periods could potentially influence the observed temporal trends and geographic differences, necessitating cautious interpretation.

## Conclusion

Our study reveals a significant increase in global ARHL prevalence cases and YLDs of among individuals aged 60 years and above from 1990 to 2021, primarily attributed to population growth. Notably, high-middle and middle SDI regions exhibited higher ASPR and ASYR, while global ASPR and ASYR demonstrated modest increases. ARHL burden is unevenly distributed, with lower SDI regions bearing a disproportionately higher burden. These findings underscore the urgent need for comprehensive strategies, including public education, occupational protections, early screening, and research into effective treatments, to address the growing global ARHL burden.

## Supporting information

**S1 Fig. Trends in the burden of ARHL across 21 regions for individuals aged 60 and older, covering the years 1990–2021.** ARHL, age-related hearing loss.
(TIF)

**S1 Table. The number and age-standardized rate of prevalence and YLDs of ARHL at the national levels in 1990 and 2021.**
(XLSX)

## Acknowledgments

The authors express their gratitude to the Global Burden of Disease Study 2021 for providing open access to the database.

## Author contributions

**Conceptualization:** Peng Zhou, Kui Xu, Xin Pan, Ling Li.

**Data curation:** Peng Zhou.

**Funding acquisition:** Ling Li.

**Investigation:** Huiqin Wu.

**Methodology:** Huiqin Wu, Kui Xu, Xin Pan, Ling Li.

**Validation:** Kui Xu, Ling Li.

**Visualization:** Huiqin Wu, Xin Pan.

**Writing – original draft:** Peng Zhou.

**Writing – review & editing:** Peng Zhou, Ling Li.

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
