## [Decision Letter · Decision Letter 0]

10 Jul 2025

Dear Dr. Li,

Thank you for submitting your manuscript to PLOS ONE. After careful consideration, we feel that it has merit but does not fully meet PLOS ONE’s publication criteria as it currently stands. Therefore, we invite you to submit a revised version of the manuscript that addresses the points raised during the review process.

**As suggested by the reviewers, please highlight the novelty of the present results.**

We look forward to receiving your revised manuscript.

Kind regards,

Paul H Delano, Ph.D.

Academic Editor

PLOS ONE

**Journal Requirements:**

1. When submitting your revision, we need you to address these additional requirements. Please ensure that your manuscript meets PLOS ONE's style requirements, including those for file naming. The PLOS ONE style templates can be found at https://journals.plos.org/plosone/s/file?id=wjVg/PLOSOne_formatting_sample_main_body.pdf and https://journals.plos.org/plosone/s/file?id=ba62/PLOSOne_formatting_sample_title_authors_affiliations.pdf 2. We note that the grant information you provided in the ‘Funding Information’ and ‘Financial Disclosure’ sections do not match.  When you resubmit, please ensure that you provide the correct grant numbers for the awards you received for your study in the ‘Funding Information’ section. 3. Your ethics statement should only appear in the Methods section of your manuscript. If your ethics statement is written in any section besides the Methods, please move it to the Methods section and delete it from any other section. Please ensure that your ethics statement is included in your manuscript, as the ethics statement entered into the online submission form will not be published alongside your manuscript. 4. We note that Figures 2 and S2 in your submission contain map images which may be copyrighted. All PLOS content is published under the Creative Commons Attribution License (CC BY 4.0), which means that the manuscript, images, and Supporting Information files will be freely available online, and any third party is permitted to access, download, copy, distribute, and use these materials in any way, even commercially, with proper attribution. For these reasons, we cannot publish previously copyrighted maps or satellite images created using proprietary data, such as Google software (Google Maps, Street View, and Earth). For more information, see our copyright guidelines: http://journals.plos.org/plosone/s/licenses-and-copyright. We require you to either present written permission from the copyright holder to publish these figures specifically under the CC BY 4.0 license, or remove the figures from your submission: a. You may seek permission from the original copyright holder of Figures 2 and S2 to publish the content specifically under the CC BY 4.0 license.   We recommend that you contact the original copyright holder with the Content Permission Form (http://journals.plos.org/plosone/s/file?id=7c09/content-permission-form.pdf) and the following text:“I request permission for the open-access journal PLOS ONE to publish XXX under the Creative Commons Attribution License (CCAL) CC BY 4.0 (http://creativecommons.org/licenses/by/4.0/). Please be aware that this license allows unrestricted use and distribution, even commercially, by third parties. Please reply and provide explicit written permission to publish XXX under a CC BY license and complete the attached form.” Please upload the completed Content Permission Form or other proof of granted permissions as an "Other" file with your submission. In the figure caption of the copyrighted figure, please include the following text: “Reprinted from [ref] under a CC BY license, with permission from [name of publisher], original copyright [original copyright year].” b. If you are unable to obtain permission from the original copyright holder to publish these figures under the CC BY 4.0 license or if the copyright holder’s requirements are incompatible with the CC BY 4.0 license, please either i) remove the figure or ii) supply a replacement figure that complies with the CC BY 4.0 license. Please check copyright information on all replacement figures and update the figure caption with source information. If applicable, please specify in the figure caption text when a figure is similar but not identical to the original image and is therefore for illustrative purposes only.The following resources for replacing copyrighted map figures may be helpful: USGS National Map Viewer (public domain): http://viewer.nationalmap.gov/viewer/The Gateway to Astronaut Photography of Earth (public domain): http://eol.jsc.nasa.gov/sseop/clickmap/Maps at the CIA (public domain): https://www.cia.gov/library/publications/the-world-factbook/index.html and https://www.cia.gov/library/publications/cia-maps-publications/index.htmlNASA Earth Observatory (public domain): http://earthobservatory.nasa.gov/Landsat: http://landsat.visibleearth.nasa.gov/USGS EROS (Earth Resources Observatory and Science (EROS) Center) (public domain): http://eros.usgs.gov/#Natural Earth (public domain): http://www.naturalearthdata.com/

Reviewers' comments:

Reviewer's Responses to Questions

**Comments to the Author**

1. Is the manuscript technically sound, and do the data support the conclusions?

Reviewer #1: Yes

Reviewer #2: Yes

2. Has the statistical analysis been performed appropriately and rigorously?

Reviewer #1: Yes

Reviewer #2: Yes

3. Have the authors made all data underlying the findings in their manuscript fully available?

Reviewer #1: Yes

Reviewer #2: Yes

4. Is the manuscript presented in an intelligible fashion and written in standard English?

Reviewer #1: Yes

Reviewer #2: Yes

**Reviewer #1**:  Dear authors

I read your manuscript with interest. The study addresses a relevant topic from a public health perspective, and the chosen methodology appears appropriate. The statistical tools employed are suitable and consistent with those commonly used in studies based on Global Burden of Disease (GBD) data.

While the manuscript is easy to read and conveys the main message clearly, I believe it is essential to explicitly clarify the substantive differences between your study and the recently published article. 'Global trends and burden of age-related hearing loss: A 32-Year Study' (Archives of Gerontology and Geriatrics, 2025, https://doi.org/10.1016/j.archger.2025.105847).

Both studies share similar objectives, data sources and methodological approaches. Your manuscript includes some differentiating elements, such as the use of YLDs instead of DALYs and the analysis of the inequality slope index, but these differences should be elaborated on more and contextualised in order to highlight their added value.

Below, I present a series of specific suggestions for improvement:

• Introduction:

A more detailed description of the Global Burden of Disease (GBD) initiative would be helpful, including its purpose, scope, and the mechanisms by which data are collected, standardised, and modelled.

With regard to age-related hearing loss (ARHL) specifically, I recommend providing a brief explanation of how the estimates are generated, including the sources used (e.g. national health surveys, clinical records, scientific literature) and the operational definitions of hearing loss employed.

• Methods:

While the manuscript adequately describes the methods used, I recommend explicitly stating which statistical software or packages were employed for each type of analysis (e.g. linear regression, Bayesian models and inequality analysis). This would improve transparency and facilitate reproducibility for other researchers.

• Discussion

It would be beneficial to strengthen the connection between your findings and their potential implications for public policy and health planning, especially considering regional disparities according to development levels.

Additionally, I suggest explicitly referencing the findings from the previously cited published study, and elaborating further on how your results complement, nuance, or challenge those findings.

You may wish to briefly discuss the limitations of relying exclusively on modeled secondary data, as is the case with GBD, and the potential implications this has for the precision of regional estimates—particularly in countries with limited availability of primary data.

**Reviewer #2:**  The manuscript by Li et al. presents well written analysis of the global burden of age-related hearing loss (ARHL) from 1990 to 2021, with projections extending to 2050. Based on data from the GBD study, the authors provide evidence that ARHL and the associated burden are increasing globally, with significant differences among regions and development status. The finding are consistent with the information and projections from the World Health Organization and contribute valuable insights for global hearing health in people > 60 years.

Overall, methodology is clear and could be replicated.

Comments

• Ln 47-48: reference for this line (7) is about employment not about cognitive decline. Also, the idea “…accelerates cognitive decline” could be interpreted as an overstatement.

• The information contained in Table 1 could be divided into two tables for clearer presentation (Prevalence and YLD).

• Ln 83: may be useful for the readers to have de detail of the direction of change in EAPC (eg. greater than 0 = increase).

• Figure 2: It may be helpful to choose an alternative color palette, considering the challenges that red–green combinations pose for individuals with color vision deficiency. A useful reference could be the following work: Guo, Z., Ji, W., Song, P. et al. Global, regional, and national burden of hearing loss in children and adolescents, 1990–2021: a systematic analysis from the Global Burden of Disease Study 2021. BMC Public Health 24, 2521 (2024). https://doi.org/10.1186/s12889-024-20010-0

• Figure 6 shows the frontier analysis of SDI and YLDs of ARHL among individuals aged 60 and older of the 1990-2021 period. The caption of the figure mention “Frontier analysis…. in 2021”. Please check and modify.

A plot B could be added to the figure to show 2021 data, highlighting some countries with greater potential to reduce ARHL. For the dots representing individual countries, it may be useful to use different colors for increase or decrease in the effective difference. This article shows something similar: Chen, X., Zhou, C. W., Fu, Y. Y., Li, Y. Z., Chen, L., Zhang, Q. W., & Chen, Y. F. (2023). Global, regional, and national burden of chronic respiratory diseases and associated risk factors, 1990-2019: Results from the Global Burden of Disease Study 2019. Frontiers in medicine, 10, 1066804. https://doi-org.uchile.idm.oclc.org/10.3389/fmed.2023.1066804

• The focus of this work was people 60 and older, which is described as a limitation of the study by the authors (not including younger population). However, I do not consider this a major limitation, as this age group is relevant for the analysis of ARHL.

However, if this point is important for the authors, it would be helpful to support the choice of age cutoff with data from WHO and other sources to strengthen the rationale. For example, including individuals aged 50 and older.

**Do you want your identity to be public for this peer review?** For information about this choice, including consent withdrawal, please see our Privacy Policy

Reviewer #1: No

Reviewer #2: No

---

## [Author Response · Author response to Decision Letter 1]

3 Sep 2025

Response to Reviewers

Dear Editor,

We sincerely thank you and the reviewers for your time and valuable comments on our manuscript. We have carefully considered all suggestions and revised the manuscript accordingly. We believe these revisions have significantly improved the quality of our work. All changes have been highlighted in tracked changes.

Editor's Comments

We greatly appreciate the editor's careful review and helpful guidance.

Comment 1: Please ensure that your manuscript meets PLOS ONE's style requirements, including those for file naming.

Response: Thank you for this reminder. We have carefully reviewed and ensured full compliance with PLOS ONE's style requirements.

Comment 2: When you resubmit, please ensure that you provide the correct grant numbers for the awards you received for your study in the 'Funding Information' section.

Response: We note the discrepancy mentioned between the 'Funding Information' and 'Financial Disclosure' sections. During the original submission, we could only locate the 'Funding Information' section in the submission interface and entered all funding details there. To ensure consistency, our corrected funding statement is: This work is supported by the National Natural Science Foundation of China (82301789), the Program of Excellent Doctoral (Postdoctoral) of Zhongnan Hospital of Wuhan University (Grant No. ZNYB2021020), Youth Interdisciplinary Special Fund of Zhongnan Hospital of Wuhan University (Grant No. ZNQNJC2023007).

Comment 3: Your ethics statement should only appear in the Methods section of your manuscript. If your ethics statement is written in any section besides the Methods, please move it to the Methods section and delete it from any other section.

Response: Thank you for your clarification. The ethics statement has been moved exclusively to the Methods section as requested.

Comment 4: We note that Figures 2 and S2 in your submission contain map images which may be copyrighted. All PLOS content is published under the Creative Commons Attribution License (CC BY 4.0)... For these reasons, we cannot publish previously copyrighted maps or satellite images created using proprietary data, such as Google software (Google Maps, Street View, and Earth).

Response: We appreciate your careful attention to copyright compliance. We have removed Figures 2 and S2 to avoid any potential copyright issues.

Reviewer #1

We sincerely thank Reviewer #1 for the thorough and constructive review. We particularly appreciate you bringing to our attention the article "Global trends and burden of age-related hearing loss: A 32-Year Study" (Archives of Gerontology and Geriatrics, 2025, https://doi.org/10.1016/j.archger.2025.105847), which was published during our submission period. We have now addressed how our work differs from and complements this important publication.

Comment 1 - Introduction: A more detailed description of the Global Burden of Disease (GBD) initiative would be helpful, including its purpose, scope, and the mechanisms by which data are collected, standardised, and modelled. With regard to age-related hearing loss (ARHL) specifically, I recommend providing a brief explanation of how the estimates are generated, including the sources used (e.g. national health surveys, clinical records, scientific literature) and the operational definitions of hearing loss employed.

Response: We thank the reviewer for this valuable suggestion. We have added detailed GBD methodology and ARHL estimation procedures in the Introduction and Methods sections, which we believe substantially strengthens the manuscript.

Comment 2 - Methods: While the manuscript adequately describes the methods used, I recommend explicitly stating which statistical software or packages were employed for each type of analysis (e.g. linear regression, Bayesian models and inequality analysis).

Response: We agree that transparency is crucial for reproducibility. We have now added comprehensive statistical software and R package information in the Methods section.

Comment 3 - Discussion: It would be beneficial to strengthen the connection between your findings and their potential implications for public policy and health planning, especially considering regional disparities according to development levels. Additionally, I suggest explicitly referencing the findings from the previously cited published study, and elaborating further on how your results complement, nuance, or challenge those findings. You may wish to briefly discuss the limitations of relying exclusively on modeled secondary data, as is the case with GBD, and the potential implications this has for the precision of regional estimates—particularly in countries with limited availability of primary data.

Response: We greatly appreciate these insightful suggestions. The Discussion section has been substantially enhanced with:

• Strengthened policy implications and regional disparities analysis

• Explicit comparison with the 2025 recently published study

• Critical discussion of GBD modeled data limitations These additions have significantly improved the depth and practical value of our manuscript.

Reviewer #2

We are grateful to Reviewer #2 for the detailed and constructive feedback that has helped improve the clarity and accessibility of our manuscript.

Comment 1: Ln 47-48: reference for this line (7) is about employment not about cognitive decline. Also, the idea "...accelerates cognitive decline" could be interpreted as an overstatement.

Response: Thank you for catching this error. We have corrected the reference and modified the wording from "accelerates cognitive decline" to "is associated with increased risk of cognitive decline" for better accuracy.

Comment 2: The information contained in Table 1 could be divided into two tables for clearer presentation (Prevalence and YLD).

Response: We agree that this improves clarity. We have divided the content into Table 1 (Prevalence) and Table 2 (YLD). Thank you for this helpful suggestion.

Comment 3: Ln 83: may be useful for the readers to have the detail of the direction of change in EAPC (eg. greater than 0 = increase).

Response: Excellent point. We have added clear EAPC interpretation in the Methods section for reader clarity.

Comment 4: Figure 2: It may be helpful to choose an alternative color palette, considering the challenges that red–green combinations pose for individuals with color vision deficiency.

Response: We appreciate your attention to accessibility. As noted in our response to the Editor, Figure 2 and S2 has been removed due to copyright concerns.

Comment 5: A plot B could be added to the figure to show 2021 data, highlighting some countries with greater potential to reduce ARHL. For the dots representing individual countries, it may be useful to use different colors for increase or decrease in the effective difference.

Response: Thank you for this excellent suggestion. We have enhanced Figure 6 with Panel B showing 2021 data with differential coloring as recommended. Corresponding descriptive text has been added to the Results section.

Comment 6: The focus of this work was people 60 and older, which is described as a limitation of the study by the authors (not including younger population). However, I do not consider this a major limitation, as this age group is relevant for the analysis of ARHL.

Response: We appreciate your perspective on this point. Based on your feedback, we have removed this from the limitations section, as we agree that the 60+ age group focus is indeed appropriate for ARHL analysis.

Once again, we sincerely thank the editor and reviewers for their time and effort in reviewing our manuscript. We believe that the manuscript has been substantially improved through this revision process. We hope that you will find the revised version suitable for publication in PLoS One.

Sincerely,

Ling Li

Corresponding author: Ling Li, MD, PhD

Department of Nuclear Medicine, Zhongnan Hospital of Wuhan University, No.169 Donghu Road, Wuhan, 430071, China

E-mail address: linglizn21@whu.edu.cn

---

## [Decision Letter · Decision Letter 1]

22 Dec 2025

The global burden of age-related hearing loss among individuals aged 60 years and older: an analysis for the global burden of disease study 2021 and predictions to 2050

PONE-D-25-09817R1

Dear Dr. Li,

We’re pleased to inform you that your manuscript has been judged scientifically suitable for publication and will be formally accepted for publication once it meets all outstanding technical requirements.

Kind regards,

Paul H Delano, Ph.D.

Academic Editor

PLOS One

Additional Editor Comments (optional):

Reviewers' comments:

Reviewer's Responses to Questions

**Comments to the Author**

Reviewer #1: All comments have been addressed

Reviewer #2: All comments have been addressed

2. Is the manuscript technically sound, and do the data support the conclusions?

Reviewer #1: Yes

Reviewer #2: Yes

3. Has the statistical analysis been performed appropriately and rigorously?

Reviewer #1: Yes

Reviewer #2: Yes

4. Have the authors made all data underlying the findings in their manuscript fully available?

Reviewer #1: Yes

Reviewer #2: Yes

5. Is the manuscript presented in an intelligible fashion and written in standard English?

Reviewer #1: Yes

Reviewer #2: Yes

Reviewer #1: Dear Authors,

I have carefully reviewed the new version of the manuscript and appreciate the effort made to respond to the comments from the first round. In my opinion, you have adequately complied with the recommendations made. The work was already solid from a statistical and methodological point of view, and the modifications introduced reinforce this impression.

The introduction is now better contextualized, with a clearer statement of the problem and a more complete description of the GBD framework and how the estimates used are generated. Likewise, the incorporation of additional technical details improves the transparency and reproducibility of the analyses. Finally, the discussion descends more explicitly into the implications for public policy, which enhances the practical usefulness of the results.

I have recommended the manuscript for publication.

Best regards

Reviewer #2: (No Response)

**Do you want your identity to be public for this peer review?** For information about this choice, including consent withdrawal, please see our Privacy Policy

Reviewer #1: No

Reviewer #2: No

---

## [Editor Report · Acceptance letter]

PONE-D-25-09817R1

PLOS One

Dear Dr. Li,

I'm pleased to inform you that your manuscript has been deemed suitable for publication in PLOS One. Congratulations! Your manuscript is now being handed over to our production team.

Kind regards,

on behalf of

Dr. Paul H Delano

Academic Editor

PLOS One